# Non–IgE- or Mixed IgE/Non–IgE-Mediated Gastrointestinal Food Allergies in the First Years of Life: Old and New Tools for Diagnosis

**DOI:** 10.3390/nu13010226

**Published:** 2021-01-14

**Authors:** Mauro Calvani, Caterina Anania, Barbara Cuomo, Enza D’Auria, Fabio Decimo, Giovanni Cosimo Indirli, Gianluigi Marseglia, Violetta Mastrorilli, Marco Ugo Andrea Sartorio, Angelica Santoro, Elisabetta Veronelli

**Affiliations:** 1Operative Unit of Pediatrics, S. Camillo-Forlanini Hospital, 00152 Rome, Italy; 2Immunology and Allergology Unit, Department of Mother-Child, Urological Science, Sapienza University of Rome, 00161 Rome, Italy; caterina.anania@uniroma1.it; 3Operative Complex Unit of Pediatrics, Belcolle Hospital, 00100 Viterbo, Italy; cuomoba@gmail.com; 4Department of Pediatrics, Vittore Buzzi Children’s Hospital, University of Milan, 20154 Milan, Italy; enza.dauria@unimi.it (E.D.); marco.sartorio@unimi.it (M.U.A.S.); 5Department of Woman, Child and of General and Specialized Surgery, University of Campania “Luigi Vanvitelli”, 80100 Naples, Italy; fabio.decimo@unicampania.it; 6Pediatric Allergology and Immunology (SIAIP) for Regions Puglia and Basilicata, 73100 Lecce, Italy; gindirli@libero.it; 7Pediatric Clinic, Pediatrics Department, Policlinico San Matteo, University of Pavia, 27100 Pavia, Italy; giamar04@unipv.it; 8Operative Complex Unit of Pediatrics and Emergency, Giovanni XXIII Hospital, 70056 Bari, Italy; violetta.mastrorilli@libero.it; 9Pediatric Clinic, Mother-Child Department, University of Parma, 43121 Parma, Italy; angelica.santoro204@gmail.com; 10Food Allergy Committee of the Italian Society of Pediatric Allergy and Immunology (SIAIP), Pediatric Department, Garbagnate Milanese Hospital, ASST Rhodense, 70056 Garbagnate Milanese, Italy; bettyvero@libero.it

**Keywords:** non-IgE gastrointestinal food allergy, eosinophilic gastrointestinal disorders, fecal biomarkers, IgG and IgG4, allergen-specific lymphocyte stimulation test, oral food challenge, atopy patch test, clinical score, endoscopy

## Abstract

non-IgE and mixed gastrointestinal food allergies present various specific, well-characterized clinical pictures such as food protein-induced allergic proctocolitis, food protein-induced enterocolitis and food protein-induced enteropathy syndrome as well as eosinophilic gastrointestinal disorders such as eosinophilic esophagitis, allergic eosinophilic gastroenteritis and eosinophilic colitis. The aim of this article is to provide an updated review of their different clinical presentations, to suggest a correct approach to their diagnosis and to discuss the usefulness of both old and new diagnostic tools, including fecal biomarkers, atopy patch tests, endoscopy, specific IgG and IgG4 testing, allergen-specific lymphocyte stimulation test (ALST) and clinical score (CoMiss).

## 1. Introduction

Food allergy (FA) is defined as an adverse health effect arising from a specific immune response that occurs reproducibly on exposure to a given food [1]. Based on the immunological mechanism involved, FA may be further classified as (a) IgE-mediated, the most well-understood form, which is caused by immunoglobulin E (IgE) antibodies against food antigens; (b) non-IgE mediated, in which the immune response is thought to act mainly through cell-mediated mechanisms; (c) or mixed, in which both IgE-mediated and cell-mediated immunological mechanisms are involved in the reaction.

IgE-mediated FA are the most common. They are easily characterized by the presence of specific serum IgE (sIgE) or a positive skin prick test (SPT). They occur most frequently in the first years of life, giving rise to urticaria/angioedema, oral allergic syndrome, rhinitis, or acute asthma and anaphylaxis [1].

Non-IgE FA are characterized by cutaneous reactions (such as atopic dermatitis, contact dermatitis and herpetiform dermatitis), respiratory reactions (such as Heiner’s syndrome) or gastrointestinal reactions, which we will discuss in more detail below [2].

Non-IgE and mixed FA are less understood, despite their frequency: a Swedish population study showed that 36% of 118 children diagnosed with cow’s milk allergy (CMA) by an oral challenge test were negative to specific IgE and SPT for cow’s milk [3].

The diagnosis of non-IgE and mixed FA is mainly clinical and is not always easy. In contrast with IgE FA, the onset of symptoms is delayed and they may have a chronic presentation, making their association with the allergen less evident [4]. Furthermore, there is a lack of laboratory tests to assist in diagnosis. In most cases non-IgE FA are diagnosed on the basis of compatible symptoms and the demonstration that symptoms disappear once the suspected food has been eliminated and reappear when it is reintroduced [5].

Oral food challenge (OFC) is considered the gold standard for the diagnosis of IgE and non-IgE FA [6]. This complex test involves the oral administration of the suspected allergen in a controlled and standardized setting, thus requiring considerable healthcare resources (physician, nurse, hospital facilities) and family support (stress, fear). Most children with non-IgE FA do not need day care hospitalization, since they are not at risk of anaphylaxis. According to the Adverse Reactions to Food Committee of the American Academy of Allergy, Asthma & Immunology, “if a patient has a negative skin test, undetectable serum food specific IgE level, and no history of convincing symptoms of immediate FA (e.g., symptoms limited to behavioral changes or delayed/chronic gastrointestinal symptoms), gradual home introduction of the food in question may be attempted” [7]. Exceptions are made for patients with suspected food protein-induced enterocolitis syndrome (FPIES), who require hospitalized medical supervision for the OFC, as they are at risk of dehydration.

The diagnosis of some forms of non-IgE FA, and especially mixed FA, also requires an endoscopic examination to reveal any eosinophilic infiltration of gastrointestinal tissue.

Recent years have seen rising interest in non-IgE FA. A search for “non IgE mediated food allergy” on Pubmed revealed 9 articles published in 2000 and 11 in 2001, compared with 66 and 67 articles respectively for 2018 and 2019. This has resulted in a large increase in knowledge of many of its clinical and non-clinical aspects. The aim of this article is thus to provide an updated review on the different clinical pictures of non-IgE and mixed gastrointestinal FA in the first years of life. It also focuses on the role of both old and new diagnostic tools, including fecal biomarkers, atopy patch tests, endoscopy, Immunoglobulin G, Immunoglobulin G4 (IgG, IgG4), allergen specific lymphocyte stimulation test (ALST), clinical score, and other novel and future tests for the diagnosis of non-IgE and mixed FA. Celiac disease, although classified as a non-IgE-mediated food allergy, is not included in this review.

## 2. Materials and Methods

Search strategy. A comprehensive search was conducted in September 2020 using MEDLINE via PubMed (www.pubmed.gov) and Embase databases (www.embase.com). Searches were not restricted by language of publication, publication type or study design, but had to have been published in the last 10 years. However, we also considered earlier relevant studies and guidelines by looking through the references of the reviews and clinical studies published on this topic. This search found 2176 articles in PubMed and 2800 in Embase. Total non-overlapping record identified in PubMed and/or Embase for each category was 4145. Of these articles 189 were considered useful and are cited in the references. The search strategies and results in PubMed (MedLine) and EMBASE are detailed in Appendix A.

The PICO (Patient, Intervention, Comparison/Intervention and Outcome) system was used to generate questions in regard to three of the topics: endoscopy and food challenge, atopy patch test and food challenge, and clinical score and food challenge. Further details on the methods and results are given in the respective paragraphs.

## 3. Clinical Features of Non-IgE Gastrointestinal FA

Non-IgE gastrointestinal FA present with specific, well-characterized clinical pictures, such as food protein-induced allergic proctocolitis (FPIAP), food protein-induced enterocolitis syndrome (FPIES) and food protein-induced enteropathy syndrome (FPE), eosinophilic gastrointestinal disorders (EGIDs) including eosinophilic esophagitis (EoE), allergic eosinophilic gastroenteritis (AEG) and eosinophilic colitis (EC) (Table 1), or with less specific clinical pictures. The latter come with nonspecific symptoms such as repeated regurgitation, vomiting, and watery or mucous hemorrhagic diarrhea, often in combination with other symptoms such as poor growth and crying crises (colic). These are not easily distinguishable from other childhood gastrointestinal diseases or functional disorders. Furthermore, FA itself can cause gastroesophageal reflux disease (GERD).

### 3.1. Food Protein-Induced Allergic Proctocolitis (FPIAP)

Notwithstanding a lack of prevalence studies, FPIAP is believed to be the most common non-IgE FA. It manifests in the first months of life with bloody stools in an otherwise seemingly well infant, in whom other causes of bleeding (constipation and/or anal fissures, infections, inflammatory intestinal diseases) have been excluded [8]. In a recent prospective population study in the USA, FPIAP was diagnosed from the presence of bloody stools or occult blood in 163 (18%) of 903 infants over a period of 3 years, and from the presence of occult blood alone in 63 (7%) [9]. The most frequent food trigger is cow’s milk, followed by egg, soya and corn [10]. Diagnosis was most frequently made at the age of one month, and it seemed to be the most frequent cause of rectal bleeding in infants. FPIAP is believed to resolve rapidly: two studies reported that bloody stools or occult blood disappear in a few weeks, even without an elimination diet [11,12]. However, a meta-analysis by Lozinsky et al. found that an OFC was still positive in 34/47 patients (72.4%) after three months of an elimination diet and in 10/47 (21.2%) after 1 year [13].

A recent study of 257 infants with FPIAP showed even less optimistic data: only 60% of children developed tolerance in the first year of life, although 99% did so within 3 years [14].

SPT and sIgE tests for milk are usually negative. However, about 20% of children with FPIAP may show sensitization or develop IgE-mediated allergy to offending foods over time [15]. Endoscopy and rectal biopsy may prove inconclusive, with focal erythema ulceration, diffuse nodularity, or loss of vascular pattern, or they may be normal. For this reason, and because FPIAP is usually a patchy disease, multiple biopsies are necessary for diagnosis [16].

For these reasons, it has been suggested, for otherwise seemingly well infants with suspected FPIAP, to wait 2–4 weeks for spontaneous resolution without initiating an elimination diet [17,18]. If symptoms continue, an elimination diet is started; if the hematochezia stops, a specific IgE or SPT test for the suspected food may be useful. If these are negative, the food may be reintroduced at home, but if positive, an OFC is essential. If symptoms persist after elimination, other causes of rectal bleeding (fissures, infections, necrotizing enterocolitis, chronic inflammatory bowel diseases, coagulation defects, invagination, volvulus, Hirschsprung’s disease) must be excluded to enable diagnosis [19].

An elimination diet (whether for the mother, as most cases arise during breastfeeding, or the infant) usually leads to regression of symptoms within 3–5 days, although some children may require a few weeks before any improvement is seen [8]. An OFC should be performed 2–4 weeks after the regression of symptoms. The first food excluded is usually cow’s milk, or the food suspected by the mother. If this seems ineffective, other foods (egg, nuts, etc.) [14] should be excluded [20], following the same diagnostic procedure for each food. If the allergen is detected the elimination diet is usually continued until the age of 12 months. An OFC can then be proposed to assess the development of tolerance [2].

### 3.2. Food Protein-Induced Enterocolitis Syndrome (FPIES)

FPIES usually presents acutely, although a chronic form has also been described. Acute FPIES is characterized by bouts of vomiting 1–4 h after ingestion of the food responsible [21]. Repetitive emesis is associated with progressive lethargy, which may be associated with shock, dehydration and acidosis, hypotonia, and hypotension [6]. Episodes of diarrhea may occur, usually within 24 h. The severity of the symptoms often causes patients to seek emergency medical care [22]. SPT and sIgE tests for foods are usually negative. However, over 10% of patients have food-specific IgE (atypical FPIES) and associated IgE clinical features before or after the onset of FPIES [23,24].

Any food can induce FPIES, but the most common causes vary by age and location. Rice and oats have emerged as the most common triggers in the USA, followed by cow’s milk, soya, egg, fish, fruits, and vegetables [22]. In Italy and Spain, fish is the most common solid food trigger [25,26]. According to a recent international consensus, the diagnosis of FPIES requires the major criterion and at least 3 minor criteria to be met [6] (Table 2). Infants presenting with a convincing history of FPIES likely do not require challenges to confirm their initial diagnosis. If only one episode has occurred, a diagnostic OFC is strongly recommended to confirm the diagnosis. Differential diagnosis includes sepsis, necrotizing enterocolitis, anaphylaxis, FPE, intussusception, pyloric stenosis, etc. [22].

A variety of protocols have been proposed in relation to OFCs. A recent International Consensus suggests administering a dose of 0.06 to 0.6 g (usually 0.3 g) of the food protein per kilogram of body weight, in three equal doses over 30 min [6]. Lower starting doses, longer observation periods between doses, or both should be considered in patients with a history of severe reactions [27]. It is generally recommended not to exceed a total dose of 3 g of protein or 10 g of total food (100 mL of liquid) for an initial feed and to observe the patient for 4 to 6 h [28].

The severity of chronic FPIES symptoms depends on the amount of food trigger continuously present in the diet. With low doses (e.g., solid foods or food allergens in breast milk), they manifest as intermittent vomiting and/or diarrhea and failure to thrive, without dehydration or metabolic acidosis. More regular intake (e.g., formula milk) is associated with intermittent but progressive vomiting and diarrhea (occasionally with blood), sometimes with dehydration and metabolic acidosis, and in about 50% of cases failure to thrive. Vomiting should regress within 3 days of excluding the responsible food, and its reintroduction may be followed by the sudden onset of acute clinical signs of FPIES. Without a confirmation challenge, the diagnosis of chronic FPIES remains presumptive [6].

### 3.3. Food Protein-Induced Enteropathy (FPE)

The incidence of FPE is unknown, although it seems to be less common than 20 years ago, and today is rather rare. It manifests with chronic diarrhea and consequent failure to thrive in the first 9 months of life [29]. It most frequently begins in the first two months of life, some weeks after the introduction of cow’s milk to the diet. More than half of the affected infants also show vomiting, abdominal distention, poor growth, and lack of appetite [30]. In a minority of cases it leads to iron deficiency anemia, associated with the presence of occult blood in the stool. FPE causes a malabsorption similar to that of celiac disease, with which it has been found in association. It is usually caused by cow’s milk, but may be due to soya or egg. Elimination of the responsible food leads to the regression of symptoms within 1–4 weeks, while the patchy villous atrophy it causes regresses several months after apparent clinical healing [29]. It is diagnosed by the reappearance of symptoms following the reintroduction of the food after 1–2 months [30].

### 3.4. Eosinophilic Gastrointestinal Disorders (EGIDs)

EGIDs are chronic diseases characterized by a range of gastrointestinal symptoms, eosinophilic infiltration of the gastrointestinal tract and, sometimes, peripheral eosinophilia. Diagnosis requires the exclusion of other causes of eosinophilic infiltration and the involvement of other organs.

#### 3.4.1. Eosinophilic Esophagitis (EoE)

EoE may Start in the First Years of Life. In a multicentric study of 705 patients with EoE, about half were under 11 years of age. In this subgroup the median age of diagnosis was 3 years and median age of the onset of symptoms 1.1 years (interquartile range 0.4–3 years) [31]. In the early years of life EoE presents as GERD, and it is thought to be responsible for about 10% of cases of infants requiring treatment for GERD. The clinical picture includes regurgitation, vomiting, sometimes rumination, lack of appetite, burning, and pain, causing crying after feeding and sometimes immediately after starting to feed. This leads to refusal of food and sometimes abnormal posturing of the head and neck and severe arching of the spine, associated with melena and iron deficiency anemia (Sandifer syndrome) [32]. In these cases, failure to respond to proton pump inhibitor (PPI) should increase the suspicion of EoE.

The most common symptoms of EoE, such as dysphagia and food impaction, increase with age and are more common during adolescence. Concomitant atopic conditions should increase the suspicion of EoE [33]. It is diagnosed on the basis of symptoms of esophageal disfunction and >15 eosinophils per high power field (eos/hpf) on esophageal biopsy [34,35]. Other non-EoE disorders that cause or potentially contribute to esophageal eosinophilia should be excluded. An esophageal biopsy is necessary not only for diagnosis but also to monitor the results of treatment. The endoscopic signs of EoE include esophageal rings, longitudinal furrows, exudates, edema, strictures, or narrow caliber esophagus [2] Even in the absence of macroscopic lesions, multiple biopsies are needed for diagnosis: at least 4 biopsies (2 in the proximal and 2 in the distal esophagus) according to Dellon [36], and at least six biopsies from two different sites (typically the distal and proximal esophagus) according to Liacouras [37].

Food can be a trigger for children with EoE, especially in the early years of life. Studies showed that about 70% of children were allergic to one or more food, above all cow’s milk, egg, wheat and soya. The younger the age, the more foods may be responsible [38,39].

#### 3.4.2. Allergic Eosinophilic Gastroenteritis (AEG)

AEG is much less common than EoE. It affect both adults and children, and is rarely seen in the first year of life [40,41]. In young children it may cause abdominal pain, irritability, easy satiety, vomiting, diarrhea, weight loss, anemia and hypoalbuminemia, due to protein-losing enteropathy. However, symptoms are dependent not only on the patient’s age but also on the organ affected, as well as the extent (invasion through bowel wall layers) [42]. Multiple food allergens are often implicated in this condition [42]. Peripheral eosinophilia is found in approximately 50% of patients with AEG. Serum tests for food-specific IgE antibodies or SPT reveal a food trigger in less than 50% of cases [43]. The diagnosis is made following endoscopic examination of the upper gastrointestinal tract, showing hyperemic edema and plaque in more than 50% of cases and, less frequently, erosion and ulceration [44]. The hallmark of AEG is marked eosinophilic infiltration of the gastric and/or duodenal mucosa, amounting to at least 30 eos/hpf [45]. Before initiating treatment of any AEG eosinophilic gastroenteritis, it is imperative to conduct a differential diagnosis to exclude other causes of hypereosinophilia with GI localization.

#### 3.4.3. Eosinophilic Colitis (EC)

EC is the least common form of EGIDs [46], although like the other forms, its overall frequency seems to be increasing [43]. It is usually seen in adolescents in association with inflammatory bowel disease and/or celiac disease, and more rarely in infants associated with other atopic conditions and FA [47]. Its association with FA is unclear, but probably drops with increasing age. In a retrospective study of 69 children with colonic eosinophilia, Pensabene found that FA accounted for 10%, inflammatory bowel disease 32%, irritable bowel syndrome 33%, and other diagnoses 25% of cases of EC. In another retrospective study of 49 children aged over 3 years, Yang found sIgE for cow’s milk and egg in 59.2% of cases. Elemental formula, simple elimination diet or combination therapy resulted in clinical improvement in 75%, 88.2% and 80% of patients, respectively [48].

Even if some studies found that more than half of cases of EC coexist with an allergy to cow’s milk protein, soya, or peanuts, the elimination diet is not usually sufficient to treat it [47]. The IgE concentration associated with allergen stimulation does not reflect the tissue concentration at the location of the ongoing allergic inflammation. This suggests that most of the eosinophilic inflammation in the colon is associated with an IgE-independent mechanism [49]. The diagnosis is established by the presence of an increased eosinophilic infiltrate in the colon wall in symptomatic patients. However, this is problematic, as different studies have found different numbers of eos/hpf in healthy children, as well as a decrease in eos/hpf moving further down the colon [50]. Hurrel et al. suggested that more than 60 eos/10 hpfs in the lamina propria and eosinophilic infiltration in the epithelium or the muscularis mucosae are suggestive of eosinophilic proctocolitis [51]. However, there is no consensus on what comprises pathologic colonic eosinophilia versus normal variation in eosinophil levels [52].

### 3.5. Less Specific Clinical Features, Other Phenotypes and Associations

Non-IgE FA occur with less specific symptoms. These can include repeated regurgitation, vomiting, crises of crying, gas, poor growth, constipation or diarrhea, and it is not always possible to frame them in one of the clinical pictures listed above. There also seem to be some differences in the clinical features and laboratory findings in different ethnic groups and geographical regions [53,54]. In addition, many of these symptoms are also present in GERD, gastrointestinal functional disorders (which are much more common in the first year of life), and irritable bowel disease. Diagnosis is thus particularly difficult in these cases, not least because the different conditions can coexist in the same child. A relationship has been hypothesized between GERD and FA, in particular CMA [55,56]. According to Nielsen, 56% of children with severe GERD may also have CMA [57]. The same association (with percentages ranging from 16 to 55%) was also found in other studies [58,59]. In addition, response to diet may not help in the diagnosis of CMA, as extensive hydrolysis can also improve symptoms in functional disorders or GERD regardless of allergy: gastric emptying time is lower in children fed with extensive hydrolytes than in those fed with adapted milk [60]. 

## 4. Fecal Biomarkers

Non-IgE FA are characterized by intestinal inflammation and increased permeability, which leads to migration of granulocytes and eosinophils to the intestinal lumen. Due to the lack of reliable diagnostic tests, there is growing interest in finding fecal biomarkers. Several studies have investigated the use of various fecal biomarkers for diagnosis, such as fecal calprotectin (FC), α-1 antitrypsin (AT), β-defensin, tumor necrosis factor-α (TNF-α), fecal IgA, eosinophil-derived neurotoxin (EDN) and eosinophilic cationic protein (ECP).

### 4.1. FC in the Diagnosis of CMA

FC is an S-100 group cytosolic protein. This group comprises calcium- and zinc-binding proteins, thereby depriving microorganisms of these trace elements and inhibiting some zinc-dependent enzymes [61]. FC is immunomodulatory, antimicrobic, and antiproliferative and is present in the cytoplasm of neutrophils, in the membranes of macrophages, in activated monocytes and in mucosal epithelial cells [62]. It is a non-invasive marker of gastrointestinal inflammation, as its release into the intestine is correlated with the movement of neutrophils and mononuclear cells through the intestinal wall and their turnover and migration into the intestinal lumen [63,64]. Its concentration is correlated with the level of intestinal mucosal inflammation, as confirmed by endoscopic and histological examinations of intestinal inflammatory conditions [65,66,67]. It has been in use for several years in both the follow-up and remission monitoring of subjects with chronic intestinal conditions [68].

Other intestinal proteins (AT, β-defensin, TNF-α, fecal IgA, EDN and ECP), as well as FC, have been studied in non-IgE FA, offering surrogate markers of the cellular response.

To investigate the use of FC in FA (particularly CMA) in infants, we selected 6 studies conducted in children with non-IgE FA and 3 studies conducted in a population with both IgE-mediated and non-IgE-mediated forms (Table 3 and Table 4). As was reported in a recent systematic review, some studies evaluated the use of FC as a biomarker for the diagnosis and monitoring of CMA, while several others investigated its use as a marker of intestinal response to OFC [69].

Baldassarre et al. reported significantly higher FC values in patients than controls, with values dropping by 50% after the elimination diet [70].

Beser et al. enrolled 32 infants under two years of age diagnosed with IgE and non-IgE mediated CMA by OFC. They found higher FC levels in the non-IgE mediated group, suggesting a possible use for this biomarker in the diagnosis and monitoring of non-IgE mediated gastrointestinal forms of CMA [71].

A prospective study conducted by Trillo Belizon et al. reported statistically higher FC values in infants aged 1 to 12 months diagnosed with non-IgE mediated CMA than in both infants for whom such a diagnosis had been excluded and healthy controls. Furthermore, there was a progressive decline in FC values after one to three months of a cow’s milk elimination diet, with significant differences in patients with gastrointestinal symptoms such as diarrhea or rectal bleeding. The authors concluded that an FC value <138 μg/g permits the exclusion of a diagnosis of non-IgE mediated CMA, with a sensitivity of 95% and a specificity of 78.57% [72].

In a study of 46 children with allergic colitis suggestive of CMA, Lendvai-Emmert et al. found considerably lower FC values in children who had followed a strict cow’s milk elimination diet for 3 months compared to their value at diagnosis, thereby indicating FC as a useful parameter for the diagnosis of CMA [73].

Prikhodchenko et al. monitored FC values in 18 children with FPE and 20 healthy age-matched children over the course of the disease. The mean FC concentration was higher in children with FPE than in the control group, but dropped significantly during the course of the disease. The authors concluded that FC shows promise for monitoring the course of FPE and evaluating treatment efficacy in children with FPE [74].

However, other studies reported results conflicting with those cited above. Some studies did not find any statistically significant difference between FC values on diagnosis of CMA and after a normal diet without cow’s milk, or in comparison with healthy controls [75,76].

The use of FC as a biomarker of intestinal response to OFC has been investigated in very few studies. Berni-Canani et al. investigated the presence of subclinical intestinal inflammation in response to challenge testing of an amino acid-based formula under study in 60 infants aged ≤4 years with both IgE and non-IgE mediated CMA. FC and ECP were measured both before and 7 and 14 days after the challenge. Their values were unchanged in all patients, thereby demonstrating their optimal clinical tolerance of the formula [77].

Merras-Salmio et al. found higher FC values in patients following an elimination diet with a positive challenge than in patients with a negative challenge (39 children) towards cow’s milk protein or in the controls (22 children), demonstrating the presence of mild inflammation of the intestinal mucosa during the challenge. The Mann-Whitney *p* values were significantly different between geometric means of FC values in non-IgE-mediated forms in comparison with IgE-mediated forms (18% versus 15%).

### 4.2. Other Fecal Biomarkers

Fecal biomarkers can indicate the degree of intestinal inflammation. Quantification of fecal eosinophils, above all EDN and ECP, reveals the extent of eosinophilic gastrointestinal inflammation, thus making them a non-invasive clinical biomarker. The feces of patients with FPIES, FPE and FPIAP show high levels of EDN, which remain stable at room temperature for at least 7 days, with matching histologic evidence of eosinophilic allergic colitis [78]. Kalach et al. determined various fecal markers (AT, TNF-α, β-defensin 2, secretory IgA, EDN and FC) and analyzed fecal microbiota and intestinal permeability in infants with digestive and non-digestive symptoms of CMA. A cow’s milk challenge was performed in all children after an elimination diet, with a positive result in 11 patients. Eight patients presented non-IgE mediated CMA. The EDN cut-off level of 2818 ng/g gave a sensitivity of 55% and a specificity of 71% and the authors concluded that measurement of EDN in a single spot sample is promising in the diagnosis of non-IgE CMA [79].

In a child with FPIES, Wada et al. found an increase in TNF-α following sequential measurement prior to hydrolysate challenge and a paradoxical reduction during the challenge. After the challenge, it rose again for one month. Similar results were observed with fecal IgA, which dropped during the challenge, whilst fecal EDN rose during the challenge. The authors concluded that the sequential measurement of fecal TNF- α, together with other markers of intestinal inflammation, could offer a sensitive and non-invasive method to evaluate non-IgE mediated forms of CMA [80].

In a more recent study of eight patients with FPIES and 12 age-matched healthy infants, Wada et al. determined FC, EDN and fecal IgA levels before and after the OFC, finding a significant increase in all three fecal biomarkers in all patients after ingestion of the causative food. However, FC and fecal IgA levels were much lower than EDN, and the authors suggest that fecal EDN testing after ingestion of the causative food may serve as a useful diagnostic marker of FPIES [81].

Very recently, Rycyk et al. measured simultaneous FC, EDN and TNFα in 34 infants with gastrointestinal bleeding and 25 control group infants with functional gastrointestinal disorders. FPIAP was diagnosed by open OFC in 27 infants, and the offending food was identified as cow’s milk in 23 and hen’s eggs in 4 patients. Children with FA demonstrated significantly higher FC and EDN levels than the controls (*p* < 0.05). The authors found the best diagnostic performance in a combination of FC and EDN (88.9% and 84%) respectively and concluded that FC and EDN are reliable tools in differentiating between FPIAP and gastrointestinal functional disorders in infants [82].

Finally, a prospective case-control study carried out in Chile reported a sensitivity of 84%, a specificity of 66%, a positive predictive value (PPV) of 68% and a negative predictive value (NPV) of 83% for occult blood in the diagnosis of FPIAP [83].

In conclusion, the results from the available literature do not permit us to make any recommendations concerning the use of FC in the diagnosis of non-IgE FA. Further studies are necessary, involving an adequate number of participants with uniform characteristics such as age, nutrition, and duration of elimination diet, and, above all, the use of clearly defined reference values and FC cut-off times. The latter is a problem of great importance, given that whilst for adults and children over 4 years of age there is a well-defined cut-off value for FC (50 mg/kg), values in children under 4 years are considerably higher, and no cut-off values have been established for infants under one year of age [84,85,86,87,88,89]. EDN values too are higher in younger children, suggesting the activation and increased degranulation of intestinal eosinophils in this age group, given the immaturity of their epithelial barrier and reduced ability to regulate the intestinal microbiota.

## 5. IgG, IgG4, Allergen-Specific Lymphocyte Stimulation Test (ALST)

The measurement of food-specific IgG and IgG4 antibody levels is often proposed for the diagnosis of non-IgE mediated FA, but the results are currently still uncertain. We performed a search on PUBMED and Embase to establish the diagnostic usefulness of IgG and IgG4 testing in this type of allergy, particularly in pediatric age (see Table 1). None of the six articles selected confirmed the diagnostic usefulness of these tests. The lack of robust evidence leads to uncertainty over their use in childhood, [91,92] therefore this is not currently recommended [8,90].

Stapel et al. [93] pointed out that the presence of specific IgG and IgG4 antibodies against a given food is merely an indicator of the immune system’s physiological response to repeated exposure to its components and a condition of immune tolerance, and it is logical to expect positive test results for specific IgG antibodies against food in healthy adults and children. Furthermore, the Canadian Society of Allergy and Clinical Immunology (CSACI) recently issued a position statement, in agreement with the American Academy of Allergy, Asthma and Immunology (AAAAI) and the European Academy of Allergy and Clinical Immunology (EAACI), declaring that there is no evidence of the usefulness of the IgG or IgG4 assay in identifying and/or predicting the presence of adverse reactions to foods [94].

The situation for EoE could be different. Recent data have highlighted the presence of high titers of specific food IgG4 antibodies in sera and esophageal tissue biopsy specimens from adults with EoE [95,96]. The clinical significance of these results is not yet clear, nor has the applicability of these findings to pediatric EoE, or their clinical functional significance in this population, been established. To further investigate this, we selected two studies.

Schuyler AJ et al. [97] demonstrated that high sIgG4 levels to cow’s milk proteins are much more common in children with EoE than in the control group and sIgG4/sIgE ratios were often 10,000:1 or higher, with an OR > 20 to all 3 cow’s milk proteins. Rosenberg CE et al. [98] reported that esophageal IgG subclasses were increased in pediatric subjects with EoE relative to controls; with IgG4 showing a 21-fold change, independently of age and duration of disease. Although more studies are needed, these data demonstrated that high specific sIgG4 or esophageal IgG4 levels could be useful biomarkers for the diagnosis or monitoring of EoE.

The allergen-specific lymphocyte stimulation test (ALST), also called lymphocyte proliferation or transformation test, has also recently been used to improve the diagnostic work up in non-IgE FA. We selected nine articles from the PubMed and Embase search (see Appendix A).

The ALST analyzes lymphocyte proliferation and cytokine production in a culture of peripheral blood mononuclear cells after stimulation with food antigen for 3–5 days. Response is typically reported as the percentage of stimulated cells (stimulation index). Although this test has long been used in the diagnosis and research of disorders associated with immune diseases (immunodeficiencies, cancer, malnutrition, autoimmune diseases, etc.), its role in allergic diseases is still uncertain [99,100].

Two studies showed the diagnostic utility of ALST in neonates and infants with non-IgE GI symptoms after ingestion of cow’s milk formula [101,102], but further evaluation of its sensitivity and specificity is needed in a larger population.

CMA can cause functional bowel disorders, which can create difficulty in managing pediatric surgical patients who also have CMA.

Ikeda K et al. [103] examined the effect of CMA on the management of 14 pediatric surgical patients in their institute, finding that a high LST index (normal range < 300%) was an important diagnostic tool for pediatric surgeons, who are in the front line for the treatment of neonates and infants with functional bowel symptoms.

Yagi H et al. [54] evaluated the relationship between the severity of non-IgE mediated gastrointestinal FA and both clinical and laboratory findings in neonates and infants, using a new symptom severity scale (grade 1–3). All patients tested positive to at least one milk component on ALST, with the most severely affected group (Grade 3) showing significantly higher positive levels than the other groups.

Kajita N et al. [104], in a recent case report of a 7-year-old Japanese girl with FPIES to quail egg, but not to chicken egg, reported that the ALST stimulation index (cut-off value > 180%) for quail egg yolk was higher than for other antigens, suggesting that the yolk might be a major allergen in quail-egg-induced FPIES.

Overall, all these studies showed that ALST could be a useful tool in the diagnosis of non-IgE mediated gastrointestinal FA, given that it can be performed regardless of the patient’s clinical condition and hence enables early diagnosis. Nevertheless, there are a number of limitations to its use in children: the use of antigens that are not yet standardized, the significant amounts of peripheral blood necessary for the test and the relatively long culture times (5–7 days) [105].

To try to improve these limits, Yagi H et al. [106] evaluated a more rapid allergen-specific lymphocyte stimulation test (IPAST) that detects IL2 mRNA expression by quantitative reverse transcription polymerase chain reaction within 24 h, using only small amounts of blood. Peripheral blood mononuclear cells from 16 young children with non IgE-mediated gastrointestinal FA and 17 controls were incubated for 24 h with cow’s milk proteins. All antigens, and especially α-casein, significantly increased IL2RA mRNA expression in patients with non-IgE-GI FA compared to the controls, with similar results to those obtained with conventional ALSTs. The authors concluded that IPAST may be a useful alternative to ALST in the diagnosis of non-IgE-GI FA, due to its high diagnostic value, small requirements for peripheral blood and rapid analysis.

In conclusion, the possibility of using specific biomarkers in the diagnosis of non-IgE mediated FA is still uncertain. While ALST and IPAST appear very promising in this regard, further studies are needed for both tests to improve standardization, to enable their use for as many antigens as possible and to better understand the mechanisms underlying the expression of cytokines and/or their receptors.

## 6. Accuracy of Atopy Patch Test Compared to OFC

The atopy patch test (APT) is an in vivo test that aims to reproduce the allergic reaction by application of the suspected allergen to the skin. It mimics the cell-mediated immune responses in which T cells play a prominent role, such as in non-IgE FA. APT has been included as a potential test to assess suspected FA in subjects with clinical signs of FPIES, FPIAP, FPE and EGID, as well as those with less specific symptoms [107].

APT is performed by applying the suspected food allergen to healthy untreated skin [108].The diagnostic accuracy of APT has been reported as higher with fresh foods than with freeze dried food extracts [109]. Any food can be assessed with patch testing, although cow’s milk, hen’s egg, wheat, and soya have been studied most extensively.

Reactions are traditionally classified as + in the presence of erythema, slight infiltration and, possibly, papules; ++ in the event of erythema, infiltration, vesicles and papules; and +++ for intense erythema with infiltrate and coalescing vesicles. They are negative if the skin is unaffected, and doubtful in the event of faint erythema only [110].

APT is not recommended for the routine diagnosis of FA [111]. Certain factors need to be taken into consideration, such as the lack of standardized test substances and wide variability in the sensitivity and specificity of results in previous studies. Moreover, there is no consensus among experts regarding the appropriate reagents, methodology or interpretation of results. The recent EAACI FA and anaphylaxis guidelines [6] states that APT remains under study, and that to date its use has not been well established. In contrast, a systematic review for FA diagnosis published in 2014 by Sampson HA et al. [42] showed some evidence that APT may be valuable in assessing food triggers in pediatric EoE. [Strength of recommendation: Moderate; Evidence: C]

We evaluated the accuracy of APT compared to OFC using a PICO system. All eligible studies had to meet the inclusion criteria: pediatric patients, FA adequately confirmed by OFC, specific results in relation to the accuracy of APT. Data extraction was developed on the inclusion criteria, taking in consideration the best available evidence. Studies from which it was impossible to extract data on the specificity and sensitivity of APT were excluded. Two reviewers screened all abstracts and full-text articles independently. Any disagreement was resolved by a third party.

A total of 56 articles were identified by the literature search. Two of these were systematic reviews [112,113], and one a meta-analysis. As this kind of study is considered to be the highest quality evidence, we have provided an overview of the research published since then. The last comprehensive systematic review searches were conducted in September and November 2017; we continued the search up to October 2020.

Articles were screened, but no relevant studies were found in addition to those already included in the systematic reviews, and only two studies published after 2017 were eligible according to the inclusion criteria.

The methodological quality of one of the systematic reviews [113] is low. The authors include 37 studies without a specified reference list, the inclusion and exclusion criteria are not clearly indicated and they did not consider the quality of included studies. Given the lack of information on the included studies, we have not taken this review into account.

The second systematic review [112] showed a good quality assessment. The authors indicated the PICO research approach and criteria for selecting eligible studies, and they included estimates of likely bias to give quality weights. This review evaluated studies of the diagnostic value of APT compared to OFC in children with FA. A total of 41 studies were included and their quality was assessed by QUADAS-2.

Most of the included studies investigated both IgE-mediated and non-IgE FA. Subgroup analyses were conducted in relation to the patients’ age and clinical signs. The gastrointestinal symptoms analyzed were: vomiting, regurgitation, diarrhea, abdominal pain, constipation, hematochezia and failure to thrive. In other cases, the enrolled patients had a specific diagnosis of enterocolitis, enteropathy, gastroesophageal reflux (GER) or FPIES. Subgroup analyses for gastrointestinal allergic symptoms indicated high specificity (91.5%) and low sensitivity (57.4%) for APT. The results showed that FA cannot be ruled out completely in the event of a negative APT, while its high specificity means that a positive APT indicates a high risk of FA.

It should perhaps be emphasized that four studies in the above systematic review recruited patients with a non-IgE mediated reaction, a negative SPT and negative specific IgE to the suspected foods. One of these was a retrospective study on FPIES, [114] while two were prospective studies conducted on FPIES [115] and on non-IgE-mediated CMA [116]. The prospective FPIES study demonstrated excellent sensitivity and negative predictive value (both 100%), while the study of children with non-IgE-mediated CMA confirmed that caution is needed before performing an OFC in children with a positive APT, given their good specificity and PPV (respectively 88.3% and 82.8%). The last of these four studies [117] was conducted on patients with non-IgE mediated rectal bleeding. Only 6 of the 31 subjects enrolled had confirmed food allergy, and none of them had a positive APT.

Our search identified two relevant studies published after the 2017 systematic review. In July 2017, Gonzaga TA et al. [118] evaluated the accuracy of APT in predicting the development of tolerance in non IgE-mediated CMA. The APTs were prepared with powdered skimmed cow’s milk in isotonic saline solution or in petrolatum vehicle and with fresh cow’s milk. With all preparation types, APT gave more false negatives than true positives. These data demonstrate the low sensitivity of APT and its low efficacy in predicting true negative patients and, hence, the development of tolerance, but also its good specificity in identifying subjects with a high risk of allergy. Cow’s milk powder in isotonic saline solution was slightly superior to the other preparations, with 33.3% sensitivity and 96.1% specificity.

The second recent study, by Sirin Kose S et al., [119] aimed to determine the diagnostic efficacy of APT compared to OFC in 133 patients with gastrointestinal symptoms caused by cow’s milk and hen’s egg allergy. The authors retrospectively investigated APT reactions compared to OFC results. APT procedures were performed by applying fresh milk or egg white and yolk on the patient’s back. The results demonstrated high specificity but low sensitivity. In patients with milk allergy APT had a specificity of 100%, sensitivity of 9.1%, PPV of 100% and NPV of 48.7%, and in patients with egg allergy APT had a specificity of 78.6%, sensitivity of 77.0%, PPV of 47.2% and NPV of 75.0%.

The discussed systematic review and our own search excluded studies without an appropriate diagnosis of FA, namely those that did not compare APT with OFC. For this reason, EoE was not included in the list of food allergies, because the diagnosis must be confirmed by the number of eosinophils in the esophageal biopsy specimen [36].

There is little literature evidence in relation to APT and EoE, with only two studies from the same group evaluated. The first, published in 2007, is a prospective study [120], while the second is a retrospective data collection [39]. These studies calculated PPV, NPV, specificity and sensitivity for different foods that caused increased eosinophils in biopsies. No other studies investigating the accuracy of APT reported specificity and sensitivity data. The prospective study [12] found PPVs ranging from 53.8% to 94.4%, depending on the food concerned, with NPVs ranging from 59% to 98.7%. It is important to point out that the last study [121] by the same authors, published in 2020 on the diagnosis and treatment of EoE, did not include the use of APT.

In conclusion, APT can be included in the diagnostic workup because it is a safe, specific diagnostic test that could point to a possible FA, especially in children with non IgE-mediated gastrointestinal symptoms (above all FPIES, FPIAP and FPE, and probably EGID too). The predictive capacity of APT can therefore be improved by combining it with negative sIgE or SPT measurement. However, several aspects require further investigation, especially to enable the better definition and standardization of the technique.

## 7. Accuracy of Endoscopy Compared to OFC

Diagnosis of non-IgE FA in clinical practice is challenging, due to the lack of pathognomonic non-invasive laboratory tests. Many non-IgE and mixed FA such as EoE and FPIAP have typical histological findings which confirm the diagnosis and point to the best treatment [122,123]. However, endoscopy with tissue sample collection can be difficult to perform, since it requires trained staff and resources. It can also be technically difficult, particularly in the first years of life, requiring general anesthesia.

Moreover, with the exception of EoE, these investigations supply data that are not easy to interpret, and hence do not change the patient’s elimination diet, the timing of trigger food reintroduction or any strong suspicions of a different diagnosis, such as autoimmune enteropathy, tufting enteropathy, microvillus inclusion disease, or congenital disaccharide deficiencies, in the case of persistent symptoms.

We aimed to compare the diagnosis of non-IgE and mixed FA based on histology and elimination diet vs. OFC. Only a handful of studies satisfied the above-mentioned diagnostic work-up (Appendix A). After excluding repeat results from the two databases or different searches, case reports, and literature reviews, 3 articles remained.

Rectal bleeding is common in not-sick newborns and infants [124]. A recent study by Jang et al. [125] aiming to clarify the etiology of small rectal bleeding in not-sick newborns demonstrated that FPIAP is a rare cause of small rectal bleeding, while idiopathic neonatal transient colitis (INTC) is far more prevalent. All 16 patients included in the study underwent endoscopy with biopsy. A food elimination test was performed in patients who did not improve spontaneously, and when rectal bleeding resolved an OFC was performed in order to confirm the diagnosis of FPIAP. Ten patients satisfied the histological criteria for FPIAP diagnosis but only two cases were confirmed as FPIAP by food elimination and OFC. One of these presented erosions on endoscopy and 141 eos/10 hpf within the lamina propria on histology, while the other had ulcers and 260 eos/hpf. Based on these results, the authors underlined that without OFC testing, INTC is often misdiagnosed as FPIAP. When the FPIAP diagnosis is based on clinical symptoms, the misdiagnosis rate is 88%, when based on clinical and pathological guidelines it is 80%, and when based on an elimination diet it is a little lower, at 67%. Most cases proved to be INTC, which has similar clinical symptoms and histopathological findings to FPIAP but resolves spontaneously without diet avoidance or medical treatment within the first week of life (average time 4 days). The authors thus suggest the usefulness of waiting for spontaneous remission of the hematochezia, in agreement with other authors, who suggest a “one month watch and wait (W&W) approach” [8,36,39,90,91,92,93,94,95,96,97,98,99,100,101,102,103,104,105,106,107,108,109,110,111,112,113,114,115,116,117,118,119,120,121,122,123,124,125].

With persistent bleeding, a diagnosis of FPIAIP should be confirmed by avoidance diet and oral food reintroduction at home, or OFC under supervision if SPT and sIgE tests for food are positive. However, in clinical practice if symptoms disappear and the infant is well, the confirmatory oral provocation test may be overcome in the first months of life. It should be periodically performed over the first year of life to test the acquisition of tolerance.

Given the age of presentation and the favorable course in the majority of cases, biopsies are generally not recommended, except in cases of unusual or abnormal symptoms such as constipation, diarrhea with mucus-streaked stools but without grossly visible bleeding, or severe rectal bleeding complicated with anemia despite a cow’s milk elimination diet.

In EoE the problem is far more complex. EoE is a chronic esophageal inflammatory disease characterized clinically by symptoms of esophageal dysfunction and histologically by eosinophil-predominant inflammation [34]. When EoE is suspected, the first diagnostic test is upper gastrointestinal endoscopy. The role of allergy testing to identify triggering foods is limited in EoE, and such foods might only be identified by an elimination diet and reintroduction of single foods under biopsy control. Although no specific recommendations exist, it is reasonable to recommend sIgE testing prior to food reintroduction under biopsy control, due to the possible loss of tolerance during the avoidance diet [126].

The search strategy (“GERD and allergy and endoscopy and oral food challenge”) identified only one study. Yukelsen et al [59]. investigated the relationship between refractory GERD (defined as the persistence of symptoms despite PPI treatment for at least 8 weeks) and allergy in 151 patients undergoing allergy testing and OFC. Of these, 28 had positive allergy tests to cow’s milk protein and 7 to egg, and also reacted during cow’s milk and egg OFC, respectively; 30 with negative allergy tests also reacted during OFC. All of them underwent endoscopy with sample collection: six patients in the first group and four in the second were diagnosed with EoE.

These results lead to various observations. First, this study showed the existence of a relationship between GERD and allergic disease. Second, it underlined that while OFC and allergy testing can identify many patients with allergic disease, endoscopy enables the diagnosis of EoE.

The detection of eosinophilic infiltration (>15/hpf) in at least one esophageal biopsy is the diagnostic hallmark of EoE (35). The recent Joint Recommendations of the North American Society for Pediatric Gastroenterology, Hepatology, and Nutrition and the European Society for Pediatric Gastroenterology, Hepatology, and Nutrition report a diagnostic algorithm addressing gastroesophageal reflux disease (GERD) management in clinical practice [127].

Upper GI endoscopy with biopsies should be performed in cases of persistent symptoms, such as crying, vomiting, anemia, feeding problems and/or failure to thrive, to identify and characterize esophagitis or enteropathy.

Poddar [128] et al. performed sigmoidoscopy and rectal biopsy in forty children presenting with a presumptive diagnosis of CMA based on clinical history of diarrhea, response to cow’s milk withdrawal and exclusion of other disease. Aphthous ulcers were found on sigmoidoscopy, while rectal biopsy revealed eosinophilia without much change in the crypt architecture. There was a recurrence of histological lesions in all patients who underwent challenge after 6 months of exclusion diet, but only 42% were symptomatic. This study showed that the correlation between histology and clinical features can be slippery: while all the symptomatic patients had endoscopic/histologic alterations at baseline, after re-challenge there was a considerable difference between histological recurrence and clinical symptoms.

## 8. Accuracy of Clinical Score Compared to OFC

The diagnosis of CMA in the first year of life is often challenging because its presentation is non-specific, especially in non-IgE mediated and mixed forms. A clinical score, the CoMiSS (Cow’s Milk related Symptom Score), has recently been proposed. According to the authors, the CoMiSS should be used as an awareness tool to help recognize the symptoms of CMA in infants and young children [129,130].

In order to review the diagnostic performance of CoMiSS and any other clinical scores, we carried out a PUBMED and Embase search using the terms listed in Table 1. We identified 363 and 328 articles in PUBMED and in Embase respectively, of which just 27 were eligible for our purposes. We found only one clinical score (CoMiSS) applicable to FA diagnosis in the first years of life. The CoMiSS is based on the presence and severity of five items investigating general clinical signs and dermatological, gastrointestinal and respiratory symptoms (Table 5).

In the first study, Vandenplas and colleagues evaluated the diagnostic performance of CoMiSS in relation to an open challenge performed after 1 month of elimination diet (extensive hydrolysate). A total of 116 infants with symptoms compatible with mild to moderate non-IgE CMA were included. The challenge was performed in 73% and was positive in 69% of the infants. The study showed a reduction in the CoMiSS during the elimination diet, and found that it was correlated to the challenge result. The average score reduction was 8.07 points; the challenge was positive in 80% of patients in whom the score was reduced to 6.0 points or less, but only in 48% if it remained ≥7 (*p* = 0.001). A more than 50% reduction in a baseline score ≥12.0 was the best predictor of a positive challenge [131].

In a later study, Vandenplas calculated the CoMiSS in 413 infants aged ≤6 months attending for vaccinations or growth examinations, in order to define normal values in apparently healthy infants and to establish the cut-off to identify those requiring further evaluation. The median and mean scores were 3.0 and 3.7 respectively, and the 95th percentile was 9.0, while only 1.5% had a score ≥12.0. Based on these results, a panel of allergologists, pediatric gastroenterologists and Belgian general pediatricians established a CoMiSS threshold of 12 or more to consider the diagnosis of CMPA likely [132].

The same authors published data on 333 healthy infants aged <6 months, documenting mean and median CoMiSS values of 2.77 and 2.83 respectively. These values rose to 3.88 and 4.00 when the analysis was extended to infants initially excluded due to incomplete data on gender and type of diet.

While no infant in the first sample had a score ≥12, 14 infants (1.9%) in this larger cohort did [132].

In another study of 226 healthy, mostly (exclusively or partially) breastfed Polish infants of the same age, the median and average scores were 4.0 and 4.7 respectively, while the 95th percentile was 11.0. Only 11 infants (4.9%) scored ≥12.0 [133].

So, the next question was: what is the clinical usefulness of CoMiSS? In particular, what is its predictive value when compared to the OFC in non IgE mediated CMA diagnosis?

Chakrabarty and Rigley evaluated the diagnostic performance of CoMiSS in two small studies, comparing it with the outcome of the elimination diet (and, in some cases only, with OFC). Interestingly, significantly reduced values were found at the end of the observation period [134,135].

Bajerova et al. used CoMiSS to identify patients at risk of CMA. The authors suggested that lower cut-offs (threshold value = 8) would increase the sensitivity of the method in children with non-specific symptoms of milk protein allergy [136].

None of the above studies reported the number of cases with sIgE and/or SPT.

Prasad carried out a study on 83 patients aged between 0 and 24 months with symptoms suggestive of CMA. A score >12.0 was obtained in 60 patients (72%). CMA was confirmed in 70 patients by OFC (performed in only 56% of cases) or ImmunoCAP. In detail, in 78.6% of patients with CoMiSS >12.0 and in 15% of patients with a value ≤12.0, the CoMiSS showed a PPV of 93% and an NPV of 33%. According to the authors the low NPV was probably because many children were already on the elimination diet and this would have led to a reduction in the score [137].

In two Italian studies, CoMiSS was compared with response to the elimination diet [138]. In the first, the PPV and NPV for score ≥12 were 100% and 9% respectively. The second was a prospective open study that investigated 47 infants aged between 1 and 12 months (median 3 months) who were on a cow’ milk protein-free diet due to the presence of persistent gastrointestinal symptoms. A significant response to the diet, defined as a ≥50% score reduction from the baseline value and below the median of the control population, was obtained in 40% of patients. The ROC curve identified a value of 9.0 as the best cut-off to predict diet response (sensitivity 84%, specificity 85% vs. 37% and 92% with a cut-off of 12; PPV 80%, NPV 88%) [139,140].

A meta-analysis of 3 studies to investigate the usefulness of CoMiSS as a predictor of CMA as confirmed by open challenge [141] found that a low score (median 5.0) after 1 month of elimination diet was associated with a higher risk of a positive challenge test (odds ratio = 0.83). Moreover, a median score reduction from 13.0 to 5.0 after a 1-month diet was predictive of the appearance of symptoms upon the introduction of cow’s milk as confirmed by the result of the confirmatory OFC, which was positive in 69% of cases in the Nestlé Health Science study [142] and in 81% in the other two studies [143,144].

Kose and Seda evaluated the response to the elimination diet according to the CoMiSS score in 112 children diagnosed with CMA, egg allergy or both. OFC confirmed the diagnosis in 46 patients (41%), in whom the modification of the score during the 1-month elimination diet was assessed. A ≥50% reduction corresponded to a sensitivity of 83.7%, 84.6% and 87.5% for milk allergy, egg allergy and both, respectively. According to the authors, this value could be employed as a cut-off for the diagnosis of the corresponding allergies [145,146].

A very recent study evaluated 168 children with a baseline score ≥12 who were started on elimination diet for 4 weeks: children who responded to the diet also underwent the open challenge. This study has two important strengths: the large number of children enrolled and the diagnostic confirmation, in all the “responders” to the diet, by open challenge. The allergy was confirmed in 54.2% patients; the ROC curve showed that the best cut-off for CoMiSS was 12.5, which corresponded to a sensitivity of 64.8% and a specificity of 54.4%. The study also showed that some symptoms, such as skin involvement, were more frequently observed in children with confirmed allergy whose score was significantly higher. The authors therefore concluded that the systematic evaluation of symptoms associated with CoMiSS can aid the selection of infants who might benefit from an elimination diet [147,148].

A collaborative study by Belgian and Spanish authors on children aged <6 months assessed the variability of the score when calculated by a pediatrician and by parents, as well as day to day variability when evaluated by parents over 3 consecutive days. The data suggest that CoMiSS can be calculated by parents, before medical consultation, without the need for special training. In the Spanish arm of the study, the diagnostic performance of the score was also compared in relation to OFC: 10 out of 13 children (76%) with a score ≥10 and 7 out of 8 with a score >12 were diagnosed as allergic to CM by OFC [149,150].

Finally, CoMiSS was recently included in a computer-based algorithm in which a score ≥12 increases the likelihood of the diagnosis and supports a dietary prescription for babies, whether exclusively breast-fed or not [151]. The score was also employed to evaluate the effect of hydrolyzed formula therapy in a study conducted at the Central Hospital of Tbilisi (Georgia). After 2 weeks, there was already a significant score reduction in children fed with hydrolyzed formula, along with a significant decrease in crying and regurgitation scores and a significant rise in the percentage of children with normal stool consistency [152].

In conclusion, 14 studies have evaluated the diagnostic efficacy of the CoMiSS. Of these, 9 were prospective studies, and 5 enrolled less than 50 children. Only about half the studies reported the percentage of positive specific IgE and/or SPT in the study population; in these studies, the vast majority seemed to be non-IgE FA. The diagnostic efficacy of CoMiSS compared to OFC was evaluated in 13 studies. In the 3 studies (Prasad, Armano, Salvatore) which used a CoMiSS cut-off value of >12, the PPV was between 80 and 100%. The vast majority of studies found a reduction in CoMiSS after elimination diet and that a >50% reduction in CoMiSS was predictive of a subsequent positive OFC.

Although further studies are needed to validate CoMiSS in the diagnostic workup of CMA and, possibly, other types of FA, and to define the optimal cut-off values, it can already be considered a useful tool, especially for suspected non-IgE mediated FA. As also affirmed by other authors, it should also be used to monitor response to therapeutic interventions such as the elimination diet, but at present it is not sufficient in itself to diagnose FA and cannot replace the OFC [153].

## 9. Novel and Future Diagnostic Tests for Non IgE-Mediated Food Allergy

Recent years have seen great interest in the search for biomarkers that, supported by clinical evidence, could facilitate the diagnostic path for non IgE-mediated and mixed FA. However, results have been poor [154]. Laboratory tests such as blood count, C-reactive protein (CRP), serum electrolytes and protein profile offer little help with the differential diagnosis in the presence of diseases with symptoms similar to non IgE-mediated FA (e.g., sepsis, gastroenteritis). However, testing for specific cytokines produced by cells involved in the immune response may be useful. A recent study in patients with FPIES identified TARC (thymus and activation-regulated chemokine) as a potential biomarker. TARC is produced by eosinophils when stimulated by TNFα and IL4, and it promotes the expression by Th2 cells of cytokine receptor type 4, which is involved in cell migration to the inflammation site [155]. TARC was initially proposed as a marker of severity and for treatment monitoring in atopic dermatitis [156]. It was recently reported that some patients with FPIES showed an increase in TARC about 24 h after being exposed to the trigger food, whether accidentally or during OFC. This increase only appears alongside gastrointestinal symptoms, suggesting that changes in serum TARC levels are likely linked to allergy reactions in intestinal epithelium cells [157,158]. This study is an example of how the measurement of cytokines and changes in their levels following OFC may help in the diagnosis of non IgE-mediated FA.

There is growing evidence that the microbiome contributes to the development and presentation of allergic diseases. It seems that gut dysbiosis likely precedes the development of food allergy, and the timing of dysbiosis appears to be critical [159]. Specific microbiome signatures have been observed in non-IgE food allergies, such as eosinophilic esophagitis and FPIAIP and FPIES [160]. This suggests that the microbiome may offer a simple and non-invasive diagnostic marker for these disorders [159].

Other studies have shown that activation of the innate immune response underlies the pathogenetic signs of these diseases. Mehr et al. used RNA sequencing and bioinformatic approaches to analyze whole blood from children with FPIES before OFC and during any acute reactions [161]. Patients reacting to the OFC showed an increased expression of the genes that activate monocytes, neutrophiles and their receptors, which are responsible for the observed reactions. In contrast, this was not observed in patients showing no reaction to OFC. In this case too, a better knowledge of the basic pathogenic mechanisms of delayed FA may contribute to the development of future new diagnostic techniques.

Similarly, Schouten et al. [162] found a high concentration of immunoglobulin free light chains (Ig-fLC) in patients with non IgE-mediated CMA who, in any case, showed a type I immediate clinical response. Increased Ig-fLC levels are normally found in chronic inflammatory diseases such as intestinal diseases, rheumatoid arthritis, Sjogren’s syndrome, systemic erythematous lupus and multiple sclerosis. Besides confirming that a chronic inflammatory state underlies allergy, this result may suggest the use of this immunoglobulin subpopulation for the diagnosis of non IgE-mediated CMA.

A recently proposed ALST measures interleukin 2 α-receptor mRNA expression within 24 h, using a small amount of peripheral blood. However, tests like these need further study to adapt their use to as many allergens as possible and to better understand the mechanisms underlying the expression of both cytokines and their receptors [106].

The basophil activation test (BAT) is a laboratory test for the in vitro simulation of an in vivo allergenic challenge, using cytofluorometric evaluation of basophil activation markers (CD63 and CD203c). The BAT is mainly used for IgE-mediated FA, but can also be used for non IgE-mediated allergies, albeit with a lower diagnostic accuracy; it is in fact one of the few in vitro techniques currently available for this kind of allergic reaction [163]. Indeed, basophil activation occurs not only through IgE signal transduction, but also as a consequence of non IgE-mediated reactions [164]. The main advantages of the BAT are its reliability, the small amount of peripheral blood required (1 mL), and its high specificity and sensitivity, enabling it to replace OFC for some patients (especially in the case of tests requiring high allergen dosages). Its limitations are related to possible basophil anergy, which is responsible for a lack of response in about 10% of cases; in addition, it requires specialized training and is still not commercially available. Furthermore, it must be carried out within 24 h (ideally within 4 h) of sampling and, last but not least, large scale validation is needed [165].

Some promising results are also arriving from the instrumental diagnostics field. To support the diagnosis of non IgE-FA in symptomatic individuals, a recent study proposed the use of abdominal ultrasound and Doppler imaging to evaluate intestinal vessel density (VD) [166]. The authors evaluated the VD of patients with a history of delayed food allergy and compared it with the VD observed in patients with gastroenteritis and in case controls. All patients with non IgE-FA showed thickening of the small intestinal wall and reduced peristalsis, and most also showed thickening of the mesentery and gastric wall. These findings suggest that non IgE-FA is characterized by a relatively severe involvement of the gastroenteric segment, as in the case of acute abdomen, gastroenteric perforation and Crohn’s disease.

In contrast, infectious diseases do not produce the same ultrasound evidence. Moreover, patients with delayed allergy showed a larger VD in the ileum and jejunum than did the other two groups. These parameters could therefore be used to distinguish a non IgE-mediated FA from a severe infection. This non-invasive examination is suitable for use in children, but as with all ultrasound procedures, it is operator-dependent. In any case, given the low number of cases included in the clinical study and the variation in the participants’ ages, further investigation is needed [166].

Finally, genetics could be used to identify individuals affected by or at risk of numerous disorders, including allergic diseases. A large number of genes have been identified by genome-wide association studies for food allergy [167]. Allergic diseases are the result of a complex interplay between genetic and environmental factors [168]. Epigenetic mechanisms may explain how the environment influences gene expression, modulating immune responses throughout life, especially early life [169,170].

Classical epigenetic mechanisms, including DNA methylation and histone modifications, have been shown to be involved in the development of IgE-mediated food allergies such as CMA [171,172,173]. Differences in DNA methylation in different gene pathways have been observed in children who subsequently developed an IgE-mediated food allergy [174], suggesting their possible role as potential biomarkers. Although epigenetic mechanisms have mostly been investigated in IgE-mediated food allergies, they are also likely to play a role in non-IgE mediated food allergy. The symptoms of non-IgE-mediated allergy to food proteins are mostly gastrointestinal and the pathogenetic mechanisms are probably cell-mediated [175].

In recent years, a common inflammatory pathway has been hypothesized for allergic diseases, characterized by a “type-2 inflammation” involving different cells besides the classic Th2 cells. These cells, from both the innate and adaptive systems, produce a unique Th set of so-called ‘type-2′ cytokines, the effectors of the allergic response [172,173,174,175,176]. This inflammatory pathway has been implicated in a wide range of allergic diseases, including atopic dermatitis, asthma and eosinophilic esophagitis [176].

Interestingly, a different DNA methylation profile of Th1 and Th2 cytokine genes and achievement of tolerance has been demonstrated in children with IgE mediated food allergy [177]. This suggests that epigenetic modifications may be potential biomarkers for predicting tolerance. The role of epigenetics in this field has been specifically demonstrated for IgE food allergies, but a similar effect might also be hypothesized for non-IgE food allergies. Different DNA methylation profiles were in fact recently demonstrated between patients with EoE who responded to treatment in comparison with non-responders [178], suggesting that epigenetic modifications may also be biomarkers of treatment response in some non-IgE mediated food allergies.

In addition to the classic mechanisms discussed above, other epigenetic mechanisms have also been proposed in non-IgE-mediated food allergy. It has been reported that post-transcriptional control elements such as miRNAs may be involved in the pathogenesis of non-IgE delayed cow’s milk hypersensitivity [179], suggesting the possibility that this reaction could be downregulated.

Although epigenetic research is still in its infancy, especially in the field of non-IgE mediated food allergies, it may have several promising clinical applications, ranging from prevention to early prediction of the success of a given therapeutic strategy. Finally, ongoing progress in molecular biology and omics sciences (e.g., genomics, proteomics, epigenomics, metabolomics, and metagenomics) may offer new insights into non-IgE food allergies [180].

## 10. Conclusions

Non-IgE mediated and mixed FA constitute a heterogeneous group of diseases arising through immunological mechanisms that are not yet well understood. In clinical practice, diagnosis generally relies on a compatible clinical history and the resolution of symptoms upon the elimination of the presumed triggering antigens. Diagnostic confirmation, however, requires a different approach in the different clinical pictures. An OFC or home reintroduction of food may be attempted in many cases, while some cases, endoscopy and biopsy of the affected intestinal tract is also essential for diagnosis. Promising new diagnostic tools to facilitate diagnosis are being studied, with encouraging results in some cases, such as CoMiSS, LSTs and IPAST. Further studies are still necessary to fully understand the physiopathology of these diseases and, consequently, improve their diagnosis and prognosis.

## Figures and Tables

**Table 1 nutrients-13-00226-t001:** Clinical features of Non–IgE- or Mixed IgE/non–IgE-mediated Gastrointestinal Food Allergies.

	Food Protein Induced Allergic Proctocolitis (FPIPC)	Acute Food Protein Induced Enterocolitis (FPIES)	Chronic Food Protein Induced Enterocolitis (FPIES)	Food Protein Induced Enteropathy Syndrome (FPE)	Eosinophilic Esophagitis (EoE)	Allergic Eosinophilic Gastroenteritis (AEG)	Eosinophilic Colitis (EC)
Age	First months of life	First year, often after the first intake of allergenic food	Weeks or months after the first administrations of the responsible food	First year of life	About 10% of children with GERD who need medication	First years of life to adult	First years of life to adult
Food allergy	Cow’s milk, egg, soya	Cow’s milk, soya, grains, pulses, poultry, fish, variable in different countries	Cow’s milk, soya, grains, pulses, poultry, fish, variable in different countries	Cow’s milk, soya, egg, wheat	Cow’s milk, soya, egg, wheat, peanut, walnut, fish	Cow’s milk, egg, fish and seafood, soya, nuts, wheat	Cow’s milk, egg
Food aversion	No	No	Sometimes	Sometimes	Yes	Sometimes	No
General condition	Good	Compromised	Compromised	Compromised	Good	Compromised	Compromised
Growth	Good at the beginning	Good at the beginning	Poor in 30% of cases	Poor	Sometimes poor	Poor	Sometimes poor
Vomiting	No	Immediate and repeated, 1-4 h after ingestion	Intermittent but progressive if the food is not withdrawn	More than half of cases	Yes	Yes	Sometimes
Regurgitation	No	No	No	No	Yes	Yes	No
Crying/colic/abdominal pain	No	No	No	No	Yes	Yes	Yes
Constipation	No	No	No	No	No	Sometimes	No
Watery diarrhea	No	in 20–50% after a few hours	Yes, chronic	Yes, chronic	Sometimes	Sometimes	Yes
Mucous diarrhea	Yes	No	No	No	No	Sometimes	Yes
Bloody diarrhea	Yes	No	Yes, in about 50% of cases	No	No	Sometimes	Yes
Abdominal distension	No	No	No	Yes	No	Yes	Yes
Acute symptoms	No	Yes	Only after the food is withdrawn	No	No	No	No
Fever	No	Sometimes	Only if acute onset after food withdrawal	No	No	No	No
Lethargy/Shock	No	Often	Only if acute onset after food withdrawal	No	No	No	No
Anemia	If not on a diet in severe forms	No	Sometimes	Yes	Sometimes	Sometimes	No
Hypoalbuminemia			Sometimes	Sometimes	Sometimes	Sometimes	No

No: almost never, Sometimes: less than 50%, Yes: more than 50%.

**Table 2 nutrients-13-00226-t002:** Diagnostic tools for Non–IgE- or Mixed IgE/non–IgE-mediated Gastrointestinal Food Allergies.

	Food Protein Induced Allergic Proctocolitis (FPIPC)	Acute Food Protein Induced Enterocolitis (FPIES)	Chronic Food Protein Induced Enterocolitis (FPIES)	Food Protein Induced Enteropathy Syndrome (FPE)	Eosinophilic Esophagitis (EoE)	Allergic Eosinophilic Gastroenteritis (AEG)	Eosinophilic Colitis (EC)
Skin Prick Test/Specific IgEs	Usually negative	Positive only in 10–20% (atypical forms)	Positive only in 10–20% (atypical forms)	Usually negative	Positive for food in about 15–20% but not always related to the responsible food	Positive for food but not always related to the responsible food	Positive for food but not always related to the responsible food
Patch test	Usually negative	Positive in different percentages between studies (21–84%)	Positive in different percentages between studies (21–84%)	Not known	Positive for food in about 10% but not always related to the responsible food	Positive for food but not always related to the responsible food	Not known
	Shows lymphonodular hyperplasia or aphthous ulceration. Histologic examination shows focal aggregates of eosinophils in the large intestinal epithelium, lamina propria, crypt epithelium, and muscularis mucosa	Not indicated	Not indicated	If performed, it demonstrates damage to the intestinal mucosa with villi atrophy	Required for diagnosis. Shows eosinophilic infiltration (>15 per hpf)	Required for diagnosis. Shows eosinophilic infiltration (>30 per hpf)	Required for diagnosis. Shows eosinophilic infiltration (often >40 per hpf)
Response to diet	Within a few days (3–5, < 10)	Immediate	Within 72 h	Within 1–4 weeks	Clinical within weeks, histological within months	Clinical within weeks, histological within months	Clinical within weeks, histological within months
Diagnosis	Possible gradual home introduction after 1–2 months	If it does not meet the diagnostic criteria	OFC in absence of previous acute reaction	OFC or reintroduction after 1–2 months	Clinical and histological remission	Clinical and histological remission	Clinical and histological remission

**Table 3 nutrients-13-00226-t003:** Fecal calprotectin in the diagnosis of CMA.

Author Year Ref	Study Design	Study Population and Sample Size	OFC	FC before Elimination Diet	FC after Elimination Diet	*p*	Comment
Baldassarre 2010 [70]	Prospective cohort study	30 (median age 8.57 months) with CMA 4 IgE mediated 26 non-IgE mediated vs. 32 (age-matched) healthy controls	No	325.89 *±* 152.31 vs. 131.97 *±* 37.98 *p* < 0.001	157.5 *±* 149.13	*p* < 0.001	FC useful for diagnosis and monitoring of non-Ige mediated CMA
Besęr 2014 [71]	Prospective cohort study	32 (median age 12.5 *±* 8.5 months) IgE mediated CMA 8 (median age 2.8 *±* 1.7 months) non-IgE mediated CMA vs. 39 (median age 11.5 *±* 7.6 months) healthy controls	Yes	392 *±* 209 886 *±* 278 vs. 296 *±* 94 *p* < 0.001 *p* = 0.142	218 *±* 90 359 *±* 288	*p* < 0.001 *p* = 0.025	FC useful for diagnosis and monitoring of non-IgE mediated CMA
Trillo Belizon 2016 [72]	Prospective	40 (median age 3.68 months) with non-IgE mediated CMA vs. 12 (median age 3.25 months) without non-IgE mediated CMA vs. 30 (median age 3.8 months) healthy controls	Yes	442.65 vs. 268.58 vs. 100.30 *p* < 0.0001	228.51 ° 92.78 °°	*p* < 0.001	FC < 138 µg/g rules out non-IgE mediated CMA. FC > 138 µg/g offers sensitivity 95% specificity 78.57% PPV 80.9% NPV 94%
Ataee 2018 [75]	Prospective cohort study	29 (median age 117.2 days) with non-IgE mediated CMA	No	209.1 (SD 387.9)	189.9 § (SD 382.4) 125.2 §§ (SD 105.4)	*p* = 0.741 *p* = 0.284	FC not useful for diagnosis or follow-up of CMA
Lendvai/Emmert 2018 [73]	Prospective cohort study	46 (median age 7.28 years) with CMA of which 36 following a strict diet	No	61.17 (SD 63.72) 77	68.35 (SD 74.74) 41.69 (SD 34.68)	*p* = 0.21 *p* < 0.001	FC useful parameter in diagnosing CMA
Diaz 2018 [76]	Prospective cohort study	17 (13–23 months) with non-IgE mediated CMA vs. 10 (age-matched) healthy controls	Yes	47.25 (28.80–106.10) vs. 68.4 (30.38–76.73) *p* = 1.0			FC not useful
Prikhodchenko/Russia [74]	Prospective cohort study	18 (1-2 months) non IgE mediated vs. 20 (age matched) controls	No	384.41 *±* 46.05 vs. 58.38 *±* 8.05 *p* < 0.001	186.29 *±* 14.16	*p <* 0.001	FC is the marker of intestinal inflammation in FPE and is useful for monitoring the disease course and evaluating the treatment

CMA = cow’s milk allergy; OFC = oral food challenge; FC = fecal calprotectin; FPE = food protein enteropathy; IgE = immunoglobulin E; SD = standard deviation; PPV = positive predictive value; NPV = negative predictive value. ° 1 month after diet; °° 3 months after diet, § 2 months after diet; §§ 6 months after diet.

**Table 4 nutrients-13-00226-t004:** Fecal calprotectin in OFC.

Author/Country/ Year/Ref	Study Design	Study Population and Sample Size	OFC	FC before Elimination Diet	FC after Elimination Diet		
BerniCanani/ Italy 2013 [77]	Prospective	60 (median age 37 months) with CMA 29 IgE-mediated 31 non-IgE mediated	Yes	36.3 *±* 21.6	32.5 ± 23.8 * 33.5 ± 21.6 ^		FC useful for monitoring intestinal response to OFC in IgE and non-IgE mediated CMA
Merras-Salmio/ Finland 2014 [90]	Prospective cohort study	57 (median age 8.7 months) with non-IgE mediated CMA vs. 22 (13.2 months) healthy controls	Yes	18 OFC positive 52 (33–86) vs. 39 OFC negative 28 (24–44)	60(30–122) 33(24–44)	*p* = 0.5 *p* = 0.4	FC not useful for diagnosis in non-IgE mediated CMA

FC = fecal calprotectin; OFC = oral food challenge; CMA = cow’s milk allergy; IgE = immunoglobulin E. * 7 days after OFC; ^ 14 days after OFC

**Table 5 nutrients-13-00226-t005:** **CoMiSS^®^:** Cow’s Milk-related Symptom Score.

Symptom	Score				
Crying (only considered if the child has been crying for 1 week or more, assessed by parents)	0	≤1 h/day			
	1	1 to 1.5 h/day			
	2	1.5 to 2 h/day			
	3	2 to 3 h/day			
	4	3 to 4 h/day			
	5	4 to 5 h/day			
	6	≥5 h/day			
Regurgitation	0	0 to 2 episodes/day			
	1	≥3 to ≤5 of small volumes			
	2	>5 episodes of >1 coffee spoon			
	3	>5 episodes of **±** half of the feeds in half of the feeds			
	4	Continuous regurgitations of small volumes >30 min after each feed			
	5	Regurgitation of half to complete volume of a feed in at least half of the feeds			
	6	Regurgitation of the complete feed after each feeding			
Stools (Bristol scale)	4	Type 1 and 2 (hard stools)			
	0	Type 3 and 4 (normal stools)			
	2	Type 5 (soft stools)			
	4	Type 6 (liquid stools, if unrelated to infection)			
	6	Type 7 (watery stools)			
Skin	0 to 6	Atopic eczema head-neck-trunk arms-hands-legs-feet			
		Absent	0		0
		Mild	1		1
		Moderate	2		2
		Severe	3		3
Urticaria	0 or 6	YES		NO	
		6		0	
RespiratorySymptoms	0	No respiratory symptoms			
	1	Slight symptoms			
	2	Mild symptoms			
	3	Severe symptoms			

The most relevant papers dealing with CoMiSS are listed in Table 6. CoMISS: Coe’s Milk Related Symptoms Score.

**Table 6 nutrients-13-00226-t006:** Characteristics of studies dealing with CoMiSS.

Author (Year)	Study Design	Number (Age)	Cases with ^+^ve IgE and/or SPT	CoMiSS vs. OFC	CoMiSS and Elimination Diet	Sensitivity/ Specificity PPV-NPV	Author’s Conclusions
Vandenplas (2014) [131]	Cohort	116 (2 weeks– 6 months)	sIgE>0.35 KU/L = 8% +ve SPT = 10%	OFC in 85/116 (73%) +ve in 59 (69%)	Basal score ≥12 If reduced to ≤6, 80% positivity of OFC.	ND	Score ≥12 useful for CMA diagnosis If reduction >50% with diet, high VPP for positive OFC
Chakrabarty (2017) [132]	Prospective	30 (24–136 days)	ND	OFC +ve in 8/10	Significant score reduction (from >12 to 6)	ND	Useful for early diagnosis and to monitor response to therapy
Rigley (2017) [133]	Prospective	58 (<1 year)	ND	OFC +ve in 2/2	Score reduction in all (from 16.5 to 3.4, average values)	ND	Useful for early diagnosis, may help reduce specialist consultations
Bajerova (2017) [134]	Cohort	121 (6 weeks– 1 year)	ND	OFC +ve in 11/18	Performed in 21	ND	A cut-off of 8 reached much more frequently in allergic patients, but a lower threshold could increase sensitivity
Prasad (2018) [135]	Observational Cross-sectional	83 (0–24 months)	^+^ve sIgE and/or SPT = 26/83 (31%)	Diagnosis confirmed in 70: by OFC in 56% of cases	ND	CoMiSS>12 Sens = 77% Spec = 66% PPV = 93% NPV = 33%	High PPV confirming the reliability of parameters included in CoMiSS
Armano (2017) [136]	Prospective	40 (3–41 months)	ND	OFC +ve in 40/40	38/40 score reduction >50%	Score ≥ 12 (in 17/40, 42.5%) predicted diet efficacy with 100% PPV and 9% NPV	Selection of candidate patients for diet
Salvatore (2019) [137]	Prospective	47 (1–12 months)	+ve SPT in 8/47 = 17%	OFC in 21/39 patients responsive to diet +ve in 6 (29%)	In 19/47 (40%) score reduction ≥50%	Best cut-off = 9 for response to diet: Sens = 84% Spec = 85% PPV = 80% NPV = 88%	To predict diet response in children with persistent GI symptoms.
Vandenplas (2013) [138]	Prospective /Multicentric	116 (80-64 days (median of two groups respectively)	sIgE>0.35 KU/L = 7.5% +ve SPT = 17% (rash); 10.5% (papule)	OFC in 85/116 (74%) +ve in 69%	Significant score reduction after 1 month diet	ND	CoMiSS useful for CMA diagnosis (OFC positive in 70% with score ≥12)
Vandenplas (2014) [139]	Prospective/ Multicentric	40 (3.4 months) (mean age)	SPT = 15/40 (37.5%) tested only 17 cases	OFC in 38/40 +ve in 38/40	Score significantly reduced after 1,3,6 months of diet	ND	ND
Vandenplas (2016) [140]	Prospective/ Multicentric	71 (6 months)	ND	OFC in 50/71 (70.4%) +ve in 34	After 1 month of diet, score significantly reduced in both confirmed and unconfirmed CMA (OFC not performed or negative)	ND	ND
Vandenplas (2017) [141]	Aggregate analysis of the previous 3 studies	See above	See above	See above	Both a score <5 (median) and a score reduction from 13 to 5 (median) after 1 month of diet increase likelihood of CMA (+ve OFC)	See above	See above
Kose (2018) [142]	Cohort	112 (5.6 months (mean)	sIgE and SPT +ve = 66/112 (59%).	OFC in 46/112 (41%)	Significant score reduction after 1 month of diet in infants allergic to milk, egg or both.	Score reduction after diet ≥50%: Sens = 83.7% 84.6%, 87.5% for milk, egg allergy or both respectively	Score reduction after diet ≥50% to be used for diagnosis of FA
Selbuz (2020) [143,144]	Prospective	168 (0–12 months)	+ve sIgE = 23/168 (13.8); +ve SPT = 20/168 (12%).	OFC in 154/168 (91.7%) +ve in 91/168 (54,2%)	After 4 weeks of diet, score reduced by ≥3 points in 154 (91.7%)	Cut-off 12.5: Sens = 64.8% Spec = 54.4%	Association of symptoms in CoMiSS helps in recognition of CM-allergic infants
Vandenplas (2020) [145,146]	Cohort	148 2.3 months (median) = Spanish cohort. 72 3 months (mean) = Belgian cohort.	ND	Spanish cohort: OFC in 13, score ≥10 +ve in 10/13 (76%), score>12 +ve in 7/8	ND	ND	ND
Kherkhheulidze (2017) [147]	Prospective	34/<1 year	ND	ND	Significant score reduction after 2 weeks of diet.	ND	ND

Abbreviations: CoMISS: Coe’s Milk Related Symptoms Score; ND = Not Done; +ve = positive; Sens = Sensitivity; Spec = Specificity; PPV = Positive Predictive Value; NPP = Negative Predictive Value; SPT = Skin Prick Test; OFC = Oral Food Challenge; CMA = Cow’s Milk Allergy; FA = Food Allergy; GI = gastrointestinal.

## Data Availability

Data sharing not applicable.

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
