# Peer review of "Non–IgE- or Mixed IgE/Non–IgE-Mediated Gastrointestinal Food Allergies in the First Years of Life: Old and New Tools for Diagnosis"

_nutrients, 2021, doi:10.3390/nu13010226_

Round 1
Reviewer 1 Report
This manuscript provides an updated review on the different clinical manifestations of non-IgE and mixed gastrointestinal FA in the first years of life. The topic is relevant and practical. Moreover the authors discuss exhaustively diagnostic tools, including fecal biomarkers, atopy patch tests, endoscopy, mmunoglobulins, allergen specific lymphocyte stimulation test, clinical scores, and other novel and future tests for the diagnosis of non-IgE and mixed FA.
The authors give a good review of up-to-date studies The article appears well referenced and sufficiently in depth. However, as currently written, it seems to be a bit difficult to follow and in my opinion could be more meaningfull if slightly rearranged. Many individual studies are described one by one with less focus on the general message coming from metaanalyses and current recomendations. The review would gain by highlighting in more comprehensive way the current recomentdations and conclusions at the end of each section.
Author Response
This manuscript provides an updated review on the different clinical manifestations of non-IgE and mixed gastrointestinal FA in the first years of life. The topic is relevant and practical. Moreover the authors discuss exhaustively diagnostic tools, including fecal biomarkers, atopy patch tests, endoscopy, immunoglobulins, allergen specific lymphocyte stimulation test, clinical scores, and other novel and future tests for the diagnosis of non-IgE and mixed FA. The authors give a good review of up-to-date studies The article appears well referenced and sufficiently in depth.
Response. We would like to thank the reviewer for their comments and suggestions, which have enabled us to improve the article considerably.
However, as currently written, it seems to be a bit difficult to follow and in my opinion could be more meaningfull if slightly rearranged. Many individual studies are described one by one with less focus on the general message coming from meta analyses and current recommendations. The review would gain by highlighting in more comprehensive way the current recommendations and conclusions at the end of each section.
Response. We have now revised the article extensively. Some sentences have been deleted and others simplified and merged. Several paragraphs have been shortened (Paragraph 4. Fecal Biomarkers is 10 lines less; Paragraph 5. IgG, IgG4, Allergen-specific lymphocyte stimulation test (ALST) is 28 lines less; Paragraph 6. Accuracy of atopy patch test compared to OFC is 19 lines less; Paragraph 8.0 Accuracy of Clinical Score compared to OFC is 40 lines less).

Reviewer 2 Report
The present review has well updated the clinical pictures of non-IgE and mixed gastrointestinal food allergies, and discussed in detailed the usefulness of both old and new diagnostic tools, including fecal biomarkers, atopy patch tests, endoscopy, specific IgG and IgG4 testing, allergen-specific lymphocyte stimulation test (ALST) and clinical score (CoMiss). The comments and suggestion the authors given were really reasonable but clear, and will be useful for us to consider further studies in the this field, and how to focus the key points to find the much better solution.
However, as pointed the authors, Non-IgE mediated and mixed FA constitute a heterogeneous group of diseases arising through immunological mechanisms that are not yet well understood, it might be difficult to review all of the related studies together in one manuscript and lead a suggestive conclusion. Therefore, the present manuscript looks a litter difficult to be understand and not be easy to read. If possible, I would like to focus on the key points more by reducing the volume of the manuscript.
Author Response
The present review has well updated the clinical pictures of non IgE and Mixed gastrointestinal food allergies, and discussed in detailed the usefulness of both old and new diagnostic tools, including fecal biomarkers, atopy patch tests, endoscopy, specific IgG and IgG4 testing, allergen-specific lymphocyte stimulation test (ALST) and clinical score (CoMiss). The comments and suggestion the authors given were really reasonable but clear, and will be useful for us to consider further studies in the this field, and how to focus the key points to find the much better solution.
Response. We would like to thank the reviewer for their comments and suggestions, which have enabled us to improve the article considerably.
However, as pointed the authors, Non-IgE mediated and mixed FA constitute a heterogeneous group of diseases arising through immunological mechanisms that are not yet well understood, it might be difficult to review all of the related studies together in one manuscript and lead a suggestive conclusion. Therefore, the present manuscript looks a litter difficult to be understand and not be easy to read. If possible, I would like to focus on the key points more by reducing the volume of the manuscript.
Response. We agree that discussing such a heterogeneous group of diseases together is complex. However, as they all have similar clinical presentations, we considered it useful to provide the reader with an overview of the different clinical frameworks and possible differential diagnoses. We have now revised the article extensively, deleting several sentences and simplifying and merging others. The revised version is 825 lines long, vs. 902 in the previous version.

Reviewer 3 Report
With interest, I read the manuscript nutrients-1033704. It is an excellently written monumental work based on extensive systematic literature search. It definitely deserves to be made available to the Nutrients Readership.
Still, I have some comment on how, at least in my view, to improve this great manuscript.
Comments:
Comm. 1. Table S1 (supplement)/Table 1 (Appendix, lines 940-941):
-> Comm. 1a. Those two are identical. Please, remove one of them.
-> Comm. 1b. I suggest adding an additional column showing the number of total non-overlapping record identified in Pubmed and/or Embase for each category.
-> Comm. 1c. Another column could indicate the number of observed records used in this manuscript (if not all).
Comm. 2. Tables are excellent, very informative and “packed” with huge amounts of the knowledge. Some cosmetic changes could be suggested.
-> Comm. 2a. They would, however, look better if oriented horizontally, especially Tables 3, 5, and 6.
-> Comm. 2b. Some things are heterogeneous, e.g. p-values in Table 3 (sometimes in italics, sometimes not, sometimes decimals without zero, etc. These all things should be unified/homogenized.
Comm. 3. Even if not directly related, I would suggest shortly referring somewhere to the paper PMID: 31547388 to highlight that also in the area of molecular analysis of the allergens the progress has been made.
Comm. 4. Lines 898-906. Classical epigenetic mechanisms have been shown to be involved in the pathogenesis of IgE-mediated food allergy, e.g. cow’s milk allergy, which have been suggested to possess a potential for future biomarkers (PMID: 28322581, 33086571, and 33193294). If you agree, please, discuss in this paragraph with suggestion that the same could be true in the case of non-IgE-mediated food allergy. And non-classical epigenetic mechanisms have been even already suggested to contribute to non-IgE-mediated food allergy (PMID: 31018604). Please, discuss as well.
Author Response
With interest, I read the manuscript nutrients-1033704. It is an excellently written monumental work based on extensive systematic literature search. It definitely deserves to be made available to the Nutrients Readership.
Response. We would like to thank the reviewer for their comments and suggestions, which have enabled us to improve the article considerably.
Still, I have some comment on how, at least in my view, to improve this great manuscript.
Comments:
Comm. 1. Table S1 (supplement)/Table 1 (Appendix, lines 940-941):
-> Comm. 1a. Those two are identical. Please, remove one of them.
Response. Table 1 in the Appendix has been removed.
-> Comm. 1b. I suggest adding an additional column showing the number of total non-overlapping record identified in Pubmed and/or Embase for each category.
Response. We have added a column showing the number of total non-overlapping records identified in PubMed and/or Embase for each category, as suggested.
-> Comm. 1c. Another column could indicate the number of observed records used in this manuscript (if not all).
Response. We also considered earlier relevant studies and guidelines by looking through the references of the reviews and clinical studies published on this topic. The studies found through our search therefore include some that are not reported in the article. Having had to change the table, we also entered other searches that we had not previously mentioned. For this reason, we prefer not to add another column to the table, as it is already quite complicated to read. Instead, we have added the following sentence to the methods: “This search found 2287 articles in PubMed and 2864 in Embase. Total non-overlapping record identified in PubMed and/or Embase for each category was 4318. Of these articles, 189 were considered useful and are cited in the references
Comm. 2. Tables are excellent, very informative and “packed” with huge amounts of the knowledge. Some cosmetic changes could be suggested.
-> Comm. 2a. They would, however, look better if oriented horizontally, especially Tables 3, 5, and 6.
Response. Table 2,3,6 have now been oriented horizontally.
-> Comm. 2b. Some things are heterogeneous, e.g. p-values in Table 3 (sometimes in italics, sometimes not, sometimes decimals without zero, etc. These all things should be unified/homogenized.
Response. Table 3 has been corrected.
Comm. 3. Even if not directly related, I would suggest shortly referring somewhere to the paper PMID: 31547388 to highlight that also in the area of molecular analysis of the allergens the progress has been made.
Response. The cited article has now been included in paragraph 9.0: Novel and future diagnostic tests for non IgE-mediated food allergy.
Comm. 4. Lines 898-906. Classical epigenetic mechanisms have been shown to be involved in the pathogenesis of IgE-mediated food allergy, e.g. cow’s milk allergy, which have been suggested to possess a potential for future biomarkers (PMID: 28322581, 33086571, and 33193294). If you agree, please, discuss in this paragraph with suggestion that the same could be true in the case of non-IgE-mediated food allergy. And non-classical epigenetic mechanisms have been even already suggested to contribute to non-IgE-mediated food allergy (PMID: 31018604). Please, discuss as well.
Response. This topic has been further developed, including some additional references (Lines 833-872).
